# The Experience of Emergency Nurses Caring for Patients with Mental Illness: A Qualitative Study

**DOI:** 10.3390/ijerph17228540

**Published:** 2020-11-18

**Authors:** Hsin-Ju Chou, Kai-Yu Tseng

**Affiliations:** 1Department of Emergency, Taichung Veterans General Hospital, Taichung 40705, Taiwan; jn178kimo@yahoo.com.tw; 2School of Nursing, Central Taiwan University of Science and Technology, Taichung 40601, Taiwan

**Keywords:** emergency nurses, psychiatric nursing, mental illness, care experience

## Abstract

Background: The medical burden of psychiatric disorders continues to increase and has caused a major impact on health, society, human rights, and economy in the world. Patients with mental illness have a higher ratio of emergency department visits than non-psychiatric patients. Psychiatric disorder-related emergency department care is a stress-causing factor in emergency department work. Therefore, the purpose of this study was to explore the experience of emergency department nurses in caring for patients with mental illness. Methods: A descriptive qualitative research design with purposive sampling was adopted. A total of 17 nurses working in the emergency department in central Taiwan were recruited. In-depth semi-structured interviews were conducted and thematic content analysis was performed. Results: Four themes and six sub-themes emerged that described the experiences of emergency nurse caring for patients with mental illness: (1) Mindset; (2) The predicament of psychiatric care: Violence and isolation and helplessness, and lack of therapeutic communication skills; (3) The influence of open space: insufficient safety and privacy; and (4) The educational needs of psychiatric nursing: improving cognition in psychiatric patients and changing negative thinking into positive thinking. Conclusions: The results revealed the experience of emergency nurses in caring for patients with mental illness. Emergency psychiatric nursing training related to foundational psychiatric knowledge, communication skill, concept of recovery, coping with violence restraining are needed for nurses who work in emergency departments.

## 1. Introduction

The World Health Organization (WHO) declared that the medical burden of psychiatric disorders continues to increase and has caused a major impact on health, society, human rights, and economy in the world [1]. Throughout the world, psychiatric disorders include psychological, physiological, and substance use disorders and affect all societies and age groups in all countries. Approximately 14% of the global disease burden can be attributed to these diseases. In Taiwan, the number of chronic psychiatric patients increased from 97,127 in 2007 to 125,932 in 2017, which was an increase of approximately 30% in a 10-year period [2]. In 2015, health insurance expenditure for psychiatric patients was ranked 8th in the National Health Insurance expenditure [3]. This shows that the number of psychiatric patients is increasing with each passing year and is one of the major areas of medical expenditure in Taiwan.

The emergency department is the first department where patients with emergencies seek assistance and its main functions are in resuscitation, salvaging limbs, and alleviating symptoms. Emergency department nurses provide a broad range of nursing care for patients from all age groups in an emergent and complex setting. This includes evaluation of symptoms, control and management of chronic diseases, stabilizing trauma injuries or acute attacks, and resuscitation for life-threatening conditions [4,5]. An ideal emergency department is a unique field that provides highly specialized care. High-quality emergency team collaboration can strengthen confidence in the team, help to decrease the incidence of errors, improve treatment outcomes in patients, and demonstrate efficacy in emergency resuscitation, which increases patient consultation and nursing staff job satisfaction [6]. Job satisfaction in emergency department staff does not arise only from salaries; affirmation and recognition from supervisors and peers is a more important source of job satisfaction [7].

A crowded emergency department is considered a basis for good management and sufficient resources. The public’s liberty to select the emergency department as a source of medical consultation provides for a convenient way to seek rapid medical attention [8]. However, this has resulted in the cognitive gap that the emergency department is a place to seek rapid, rather than emergent, medical attention, which results in an increase in misunderstanding and conflicts between emergency department nurses and patients [8]. Regular congestion in the emergency department, a spatially and temporally cramped work environment, disrespect, belittlement, lack of support, rude behavior, and excessive workload are factors resulting in physical and mental fatigue in emergency department nurses, affecting job satisfaction and retention rate [7,8,9]. In addition, teamwork is emphasized in the emergency department and insufficient professional skills in nurses will obstruct smooth cooperation in an emergency department team. Working alone in a task-centric scenario will result in a loss of team cohesion and cause difficulties and feelings of loneliness and helplessness [6].

In Taiwan, the average daily emergency department volume reached 19,672 persons in 2017 [2] and patients who frequently visit the emergency department are one of the factors leading to emergency department congestion. Studies have pointed out that psychiatric patients have a higher ratio of emergency department visits than non-psychiatric patients [10]. Psychiatric patients with dysthymic disorders, schizophrenia, delusional disorders, mental-, stress-, and physical-related disorders, and alcohol-related disorders are those who mainly use the psychiatric emergency department [11,12,13,14]. The main reasons for psychiatric patients to seek medical attention are dysthymic disorders, violent behavior, and alcohol dependence [12,14]. Younger patients tend to seek medical attention because of emotional factors, whereas older patients tend to seek medical attention for psychiatric disorders and dementia.

As the emergency department is the first department where psychiatric patients with acute attacks seek assistance, emergency department nursing provides management of acute psychiatric disorder attacks at an important time. However, when emergency department nurses perform emergency procedures and interact with patients, they often encounter abuse, patients requiring mandatory sedation, escapes, or threats of violence, and experience negative emotions of loneliness and helplessness. Thus, psychiatric disorder-related emergency department care is a stress-causing factor in emergency department work. If these nurses lack confidence in caring for psychiatric patients, this may affect the willingness of emergency department nurses in caring for psychiatric patients [15], which highlights the importance of appropriate nursing intervention in emergency care for psychiatric disorders [13]. Some studies focused on examining nurses when they encounter violence and suicidal patients [16,17]. However, there are few studies examining emergency department nursing care of psychiatric patients, particularly the experiences and thoughts of emergency department nurses when they care for psychiatric patients [18,19]. These studies identified that emergency department nurses work in a fast-paced and high stimulus environment where emphasizes rapid assessment and emergency care. Nurses found it difficult to provide an optimal care for patients with mental illness. Therefore, the aim of this study is to examine the experience of emergency department nurses in caring for emergency psychiatric patients.

## 2. Materials and Methods

### 2.1. Study Design

Experience is a subjective feeling and every individual has his/her unique thoughts and feelings and personal experiences cannot be measured and calculated. This requires in-depth contact with individuals and following contexts through the description and analysis of true event occurrences [20] for understanding the meaning produced [21]. A qualitative study was conducted, and semi-structured in-depth interviews were used for data collection.

### 2.2. Participants

The subjects of this study were emergency department nurses and we focused on their experiences in caring for psychiatric patients during the course of their work. Purposive sampling was employed and a total of 17 subjects agreed to participate in this study and completed the interview. The inclusion criteria were: (A) Taiwanese nurses aged 20 years and above with a valid and effective nursing license. (B) Working experience in an emergency department at a medical center in central Taiwan for at least 1 year. (C) Practical experience of emergency care for psychiatric patients. (D) Agree to participate in this study’s thematic interview after explanation and sign the study participation form. (E) Subjects who are able to describe their experience in Chinese.

### 2.3. Data Collection

In this study, we recruited emergency department nurses as volunteers from a medical center in central Taiwan using posters. After signing the study informed consent, interviews were scheduled based on timings that were convenient for the subjects. The interview sites selected were independent, quiet, and private settings comfortable to the subjects so that subjects could freely and fully express their experiences and thoughts in a focused manner without disturbance. Each interview lasted for 60 min. One of the authors was responsible for conducting interviews and is a nurse with more than 20 years of emergency psychiatric care experience. During the data collection period, three emergency department nurses were first interviewed and the interview questions was confirmed after discussion with two experts. The interview questions: (1) Please describe your experience and thoughts on working in the emergency department? (2) What are your views on patients with psychiatric disorders? Please discuss your experience of caring for psychiatric disorder patients when they sought medical attention at the emergency department? (3) What are the difficulties you face when caring for psychiatric disorder patients? (4) How do you cope when on encountering difficulties when caring for psychiatric disorder patients? (5) Do you receive assistance when caring for psychiatric disorder patients? Are there other needs? (6) Do you have other thoughts that you are willing to share with me? Some strategies were used to improve in-depth interviews. (1) using icebreaking techniques for relaxing interviewee; (2) using language and non-verbal language to establish a trusting relationship; (3) providing more time for reflection and response; (4) during the interview, respect and listening without criticism.

### 2.4. Data Analysis

In this study, thematic content analysis (TCA) proposed by Newell and Burnar was used for data analysis [22]. The six steps in TCA were used for analysis. The data collection interview was audio recorded and key points were noted for summary. After the interview, the key points were reviewed and the recording content was transcribed. The researchers repeatedly read and annotated the transcripts to immerse themselves in in the raw data. Open coding was used to categorize descriptions in the raw data that are associated with the study theme into titles or categories before the various coded meaning sets were used to generate major themes with common characteristics [22]. For example, a participant stated that “*I am scared that my words will enrage the patient. I do not possess communication techniques and do not know when and what to talk to patients about*” was named as a category code “communication is difficult”. As the categories were collected together, a similar set of category codes as “Lack of therapeutic communication techniques” was developed. Finally, the researcher returned to the transcripts and marked the text with different colors of marker pens to reflect category codes. In addition, the researchers constructed a reflexive journal to aid in self-examination of the interview process and analysis of the development of categories and themes. For increasing the trustworthiness of the study results, another author and a researcher from the same field jointly examined the interview data and analysis results for avoiding missed analysis of interview content [23]. The final analysis results were jointly examined by a qualitative expert and a clinical emergency nursing expert to achieve neutrality and confirmation of the study.

### 2.5. Ethical Considerations

This study was reviewed by the institutional review board of the university (ethical code: CE19114A). Before the interview, subjects were given the “subject instructions and consent form” and were fully informed about the aim of the study, process, data collection method, subjects’ rights, possible discomfort and management principles, and expected study benefits. At the same time, the researchers’ contact information was provided. In order to respect the willingness and privacy of the study subjects, the interview time and venue were decided by the subject. Subjects’ names were coded and deidentification was conducted.

## 3. Results

A total of 17 emergency department nurses participated in this study, of whom 4 were males and 13 were females. Their ages ranged from 26 to 50 years and the mean age was 33 years. The mean emergency department work experience was 9 years. Table 1 shows general information on the study subjects.

The experiences of emergency department nurses in caring for psychiatric patients were divided into four themes and six sub-themes (Table 2), which are discussed as follows.

### 3.1. Theme 1. Mindset

Mindset means that the stigma and labels of psychiatric disorders are common and deeply rooted in society. The public’s perception of psychiatric disorders is mostly derived from stereotypes and the media. A fixed traditional mindset causes psychiatric patients to be discriminated and shunned. Even emergency department nurses have a fixed impression when caring for psychiatric patients.

**Subject N**: *I think that psychiatric patients like to split hairs, are stubborn, and have incurable diseases.*

Due to their stereotypic knowledge and personal work experience toward psychiatric patients, most emergency department nurses mentioned that their evaluation and management of psychiatric patients may be affected when caring for them and this may even lead to delayed management. For example, the experience of subject A: *The patient kept repeating some imaginative things and I felt that his condition was not controlled, his words could not be trusted, and I was unable to continue conversing with him. If he actually had some type of problem, the condition may have been overlooked as I did not trust him.*

**Subject B**: *I feel that I do not respect psychiatric patients as I do not focus on communication techniques or other care techniques other than restraining them and injecting them with drugs.*

### 3.2. Theme 2. The Predicament of Psychiatric Care

Emergency department nurses often experience the predicament and limitations of psychiatric nursing when caring for psychiatric patients, including violent experiences and management, and communication with psychiatric patients. This theme includes two sub-themes: (1) Violence and isolation and helplessness, (2) Lack of therapeutic communication skills.

#### 3.2.1. Sub-Themes (1) Violence and Isolation and Helplessness

The stereotypes of psychiatric patients are often associated with “ticking time bombs,” “dangerous”, “aggressive”, and most psychiatric patients are experiencing acute attacks when they register at the emergency department. Emergency department nurses generally have experiences of violence from such patients. Subject H mentioned the worry and fear from being attacked:


*Actually, I am afraid to care for psychiatric patients as I do not know what will happen next. There was one time when I assisted in restraining a psychiatric patient and his head rose up when my hand crossed over and he bit my hand without releasing. Specifically, we are often exposed to a dangerous situation and may be bitten or beaten up at any time. We are scared of being attacked and also wonder why we should encounter violence?*


**Subject F**: *Once I was examining a patient for triage and routinely asked the patient’s family member about his/her recent condition. Suddenly, the patient had an attack and removed my mask, which was a threat at a very close distance and made me feel very scared.*

Due to their experiences of violence during care, emergency department nurses generally believe that they are unable to demonstrate teamwork and provide assistance during care and management of violence, which causes emergency department nurses to feel isolated and helpless. The emergency department is a workplace that particularly focuses on teamwork, and danger, threats, isolation, and helplessness indirectly affect the morale and enthusiasm of emergency department nurses.

**Subject F**: *When an emergency occurs in internal medicine or surgical emergency department patients, everybody will show teamwork and help. However, only nurses will step forward when an acute attack occurs in psychiatric patients. The security guards will arrive after support is sought and the emergency bell is pressed but they are often at the periphery or less important places while the nurses are in the front restraining the patient. Why don’t they come as quickly to complete these procedures as when carrying out intubation or CPR?*

**Subject G**: *Every time a psychiatric patient escapes, only nurses will run after them. Some security guards will only advise the patient to cooperate and won’t restrain them. When the patient needs to be restrained, it is always the nurses who will grab and restrain the patient. During the entire process, only nurses restrain the patient in order to protect the patient from harming himself/herself.*

#### 3.2.2. Sub-Themes (2) Lack of Therapeutic Communication Skills

When interacting with psychiatric patients and providing emergency management, emergency department nurses often endure silence or verbal abuse from patients treated unwillingly. Subject B mentioned that he/she felt that therapeutic communication technique is insufficient when caring for psychiatric patients during an acute attack, resulting in an inability to provide complete evaluation and appropriate care for psychiatric patients: *I am scared that my words will enrage the patient. I do not possess communication techniques and do not know when and what to talk to patients about. When I talk to patients with depression on why they want to hurt themselves, they will talk but they will say that I will not understand and start to cry. Therefore, I lack confidence.*

**Subject L**: *When I talk to psychiatric patients, it is difficult to distinguish whether what they say is true or false. For example, the patient might say that he wants to pee but you will suspect that he wants to escape. I lack patience when handling psychiatric patients. In addition, I feel that psychiatric communication techniques are difficult and I lack these skills.*

The lack of communication skills and confidence toward psychiatric patients in nurses affects their willingness to care. In addition, the lack of therapeutic communication techniques also results in inappropriate or unreasonable attitudes when responding to patients during care. For example, subject B mentioned that: *I dislike psychiatric patients as conversations sometimes end up as quarrels and they will scold profanities. I will feel why should I be scolded and tell them to shut up.*

**Subject K**: *Sometimes the patient will talk about imaginative stories but this will irritate me and I will tell him not to talk nonsense if there is nothing else. I feel that I am not willing to communicate with the patient or provide appropriate clarification.*

### 3.3. Theme 3. The Influence of Open Spaces

This theme of the influence of open spaces includes two sub-themes: insufficient safety and insufficient privacy.

#### 3.3.1. Sub-Themes (1) Insufficient Safety

The emergency department is an open space and differs from psychiatry wards that have doors to restrict access. During observation in the emergency department, some psychiatric patients will walk around freely without restraints. However, distinguishing psychiatric patients from emergency department patients is difficult, resulting in a hidden threat of psychiatric patients getting lost or escaping.

Subject J: *Access to the psychiatric ward is controlled but not the emergency department. If there is a patient in the protection room, other psychiatric patients will be left in the general observation zone. The emergency department is relatively open and there are people entering and leaving the department. It is difficult to identify who is a psychiatric patient and patients often easily leave the emergency department due to this reason.*

Subject A: *Psychiatric patients are left in front of the nursing station to facilitate observation. However, that zone is very close to the emergency department doors and the patient can easily escape if there is no family member accompanying the patient.*

#### 3.3.2. Sub-Themes (2) Insufficient Privacy

The emergency department is crowded and noisy and psychiatric patients are exposed to the curiosity or discussion of other patients or family members because of a lack of independent queues and spaces during consultation. Many nurses mentioned the privacy viewpoint.

**Subject P**: *The emergency department is an open space and beds are close to each other. Other people can hear our conversations. The emergency department is also not friendly to emotionally agitated patients who are restrained and it is very obvious when patients are tied up in an open space and family members will stare at such patients in a weird manner.*

Subject D also mentioned that psychiatric patients use the consultation queue the same as the general public and it is unavoidable that this arouses the curiosity of the public. He mentioned that: *The emergency department environment is not friendly to psychiatric patients and their consultation queue is the same as ordinary patients. Members of the public often surround these patients from registration to consultation and examination, and some members of the public might even want to take videos and photographs. I feel like this leaves no dignity to the patient.*

In addition to the lack of privacy for psychiatric patients in the emergency department, the open spaces in the emergency department are also a source of stress and frustration during the care of psychiatric patients by emergency department nurses. Subject C described that misunderstandings tend to occur with other patients or family members due to lack of understanding when nurses care and manage psychiatric patients: *Once, a psychiatric patient was preparing to get off the bed and I worried that he/she will injure him/herself or leave the emergency department. As I was a certain distance away from him/her, I shouted at him/her to sit properly and not to jump. An elderly man opposite him/her asked me why I was fierce to the patient and shouted at him/her instead of talking to him/her nicely. I felt misunderstood and wronged.*

### 3.4. Theme 4. Educational Needs of Psychiatric Nursing

This theme includes two sub-themes: Improving disease awareness and psychiatric care capabilities and changing to positive thinking.

#### 3.4.1. Sub-Themes (1) Improving Disease Awareness and Psychiatric Care Capabilities

A lack of self-confidence in caring for psychiatric patients in emergency department nurses affects their willingness to care and subjects mentioned that they hope to improve their knowledge of psychiatric disorders through on-the-job training or case discussions.

Subject F mentioned the importance of on-the-job training: *I require on-the-job training for psychiatric care, particularly communication skills. Psychiatric experts can be invited to employ teach-back methods and provide onsite simulation of psychiatric patients. Otherwise, some cases can be discussed so that the emergency department team learns how to better assist in psychiatric care of these patients.*

#### 3.4.2. Sub-Themes (2) Changing to Positive Thinking

In an emergency and complex environment, emergency department nurses care for psychiatric patients with no specific chief complaint and difficult evaluation. However, they still hope to define their nursing role in the midst of a heavy workload and empathize with psychiatric patients and their family members after self-reflection. This causes them to shift to positive thinking during care and provide professional and suitable care for psychiatric patients. Subject O mentioned that: *the emergency department is the first stop where psychiatric patients seek aid for acute attacks and the role and functions of the nurse should be performed better. Detailed explanation before medical procedures are carried out can decrease anxiety in patients and family members and caring for family members is as important as caring for psychiatric patients.*

**Subject A**: *When a psychiatric patient seeks medical attention, we often suppress his/her behavior in an expressionless manner. In actuality, psychological disorders are the same as physical illnesses. The emergency department may not have the means to care for their psychological aspect deeply, but we should spend more time to think about the kind of aid that we can provide to them. This leads me to contemplate.*

## 4. Discussion

The study results showed that emergency department nurses are affected by mindset, have negative views toward psychiatric patients, and intuitively experienced a lack of trust when caring for psychiatric patients. The biases of nurses affect their care attitudes and affect the completeness of emergency evaluation and suitability of subsequent management of psychiatric patients. These results are similar to the studies of Clarke et al., Pelleboer-Gunnink et al., and Lindström et al. [24,25,26]. The awareness and biases in triage nurses may affect the collection of consultation information, resulting in inappropriate triage decisions. During care, nurses generally felt that psychiatric patients are unstable and aggressive, have poor cooperation, and are harder to care than patients with regular physiological diseases. Therefore, psychiatric patients experience significant discrimination during consultation, which indirectly affects the integrity of medical care received.

Nurses are the frontline direct caregivers in the emergency department and are, therefore, exposed to the threat of violence. The findings of this study showed that subjects felt threatened and fearful when they experienced violence and developed negative emotions, which affected their work motivation. These results are consistent with those of previous studies [11,27,28] which showed that subjects felt negative emotions of doubtfulness, helplessness, and sadness after experiencing violence and the accompanying stress results in a lack of work motivation [27], which affects work attitudes and enthusiasm and decreases a sense of achievement in work and organizational commitment [11,29]. In addition, the results of this study provided another perspective: all subjects mentioned that in the face of aggressive behavior or restraining patients, only nurses will step forward to suppress the patient, whereas the emergency department care team do not provide immediate assistance. Although the hospital has developed standard operating procedures for violence management, the roles and functions of other colleagues are not clear, causing emergency department nurses to feel isolated and helpless.

Emergency department patients have diverse and complex conditions and congestion is a major challenge. Appropriate communication and listening can enable psychiatric patients to feel consoled and comfortable [30]. Based on the study subjects’ experiences of caring for psychiatric patients, the subjects felt that the emergency department environment is busy and there is a lack of therapeutic communication techniques. This results in irrational attitude and language responses by nurses toward patients. These results echoed those of previous studies in which emergency department nurses felt that they lack communication skills [31]. Psychiatric patients described how professional nurses may be good at resuscitation but not at communication and listening [26], showing a lack of nurse–patient communication skills and indicating that more communication technique training is warranted [31].

Our study found that the emergency department is an open space and lacks door access controls, which results in a risk of psychiatric patients getting lost or intentionally escaping. In particular, the consultation environment is noisy and lacks privacy and there is a lack of an independent queue and a quiet environment. This may increase patient stimulation and affect their emotional stability. In particular, restraining patients in an open environment may negatively affect recovery in patients in the future [32]. Researchers also pointed out that the emergency department generally lacks an independent and safe evaluation environment [33]. Therefore, ensuring the safety and privacy of psychiatric patients is difficult, resulting in stress and distress when staff and psychiatric patients interact.

The results of this study showed that emergency department nurses tend to focus on solving physiological problems in psychiatric care and that professional psychiatric nursing training or continued education is required for the management of psychiatric patients. This includes understanding various psychiatric disorders, restraining techniques, and communication skills, which will increase the confidence of emergency department nurses in identifying subtle signs in psychiatric patients and caring for such patients. Surprisingly, the results of this study also pointed out that emergency department nurses generally have negative perception toward psychiatric patients. Evaluation and management of psychiatric patients by emergency department nurses is deeply affected by their mindset. Therefore, inclusion of stereotypes and psychiatric patient recovery concepts in on-the-job training for emergency department psychiatric nurses and establishing a bias-free clinical environment is an urgent task. In addition, the results of this study revealed that nurses who worked in the emergency department experienced insecurity in their working environments. However, previous study stated that nurses’ attitudes are negatively related to job performance [34]. Therefore, further study requires exploring nurses’ work attitudes to evaluate subsequent job performance and their working experiences for patients with mental illness. Further research including nurses’ mindset, culture factors and quality of care is also needed.

This study is limited to the experiences of emergency department nurses from a medical center. Therefore, the results of this study cannot be completely generalized to emergency department nurses from other hospital grades. The subjects of this study were emergency department nurses who have worked for at least 1 year. Therefore, the results cannot be generalized to the experiences of newly enrolled emergency department nurses in caring for psychiatric patients.

## 5. Conclusions

This study provided an in-depth understanding of the experiences of emergency department nurses in caring for psychiatric patients. Emergency department nurses generally have negative perceptions and biases toward psychiatric patients and this mindset affects the evaluation and management of such patients. The open environment that lacks privacy causes patients to be exposed to external stimuli, increases the occurrence of violence, and also increases the difficulty and stress of caring for psychiatric patients in nurses. Increasing psychiatric care knowledge and skills and strengthening team responses toward violence are urgent needs for emergency department nurses. In patient safety, creating a safe hospital environment have been identified. Nurse managers should provide on-the-job training of psychiatric care for nurses working in the ED.

## Figures and Tables

**Table 1 ijerph-17-08540-t001:** Participant characteristics.

Participants	Gender	Age	Education Level	ED Working Years
A	F	34	Bachelor	11
B	F	28	Bachelor	4.5
C	M	26	Bachelor	2
D	M	28	Bachelor	5
E	F	26	Bachelor	4
F	F	36	Master	7
G	M	32	Master	8
H	F	28	Bachelor	5
I	F	29	Bachelor	7
J	F	30	Bachelor	7.5
K	M	26	Bachelor	3
L	F	47	Master	24
M	F	28	Bachelor	5
N	F	50	Bachelor	30
O	F	39	Bachelor	15
P	F	30	Bachelor	7
Q	F	39	Bachelor	14

ED: Emergency Department.

**Table 2 ijerph-17-08540-t002:** Results.

Theme	Subtheme
1. Mindset	
2. The Predicament of Psychiatric Care	(1) Violence and isolation and helplessness
	(2) Lack of therapeutic communication skills
3. The influence of Open Spaces	(1) Insufficient safety
	(2) Insufficient privacy
4. Educational Needs of Psychiatric Nursing	(1) Improving disease awareness and psychiatric care capabilities
	(2) Changing to positive thinking

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
