# Peer review of "The Experience of Emergency Nurses Caring for Patients with Mental Illness: A Qualitative Study"

_ijerph, 2020, doi:10.3390/ijerph17228540_

Round 1

Reviewer 1 Report

Ethics: did the participants sign an informed consent? Please clarify.

The data analysis can be more transparent. For example, you can give examples on each step, so that the Reader more easily can follow and understand the results. 

What did the participants talk about in the interviews?

I question whether the results are somewhat unbalanced. Did the nurses only tell negative narratives, only negative experiences, or did you, the researchers, only ask for the negative narratives? I find it strange that the nurses only experience negative narratives, as they have chosen to work in an emergency department. I ask: Is it the researchers' preconception: emergency Department nurses generally have negative perception and biases toward psychiatric patients, that is confirmed in this study? Please Balance Your results. 

Reviewer 2 Report

This article highlights a topic of major impact in society, the needs in the care of psychiatric disease and the lack of preparation to address care of these patients.

In line 17 at the beginning "was adopted. A total of 17 nurses ..." improve the sentence.

Reviewer 3 Report

Thank you for allowing me to review this manuscript. The care of psychiatric patients is an issue that must be explored. This study can be improved at a methodological level with a series of modifications.

Abstract

Lines 15-16, is a justification for the study, it should not appear in the abstract. However, the objective is not described.
Introduction
Lines 86-87. The authors report that few studies determine nursing care in the emergency services in psychiatric patients. What do these studies determine?
Materials and methods
Indicate the methodological approach of the qualitative study.
How were the in-depth interviews carried out in data collection?
Results
It would be interesting in Table 1 if the individual characteristics of each participant were described. This would allow us to identify the participants in the verbatims.
In Table 2 the subtopics corresponding to each topic are not clear.
Discussion
Study limitations should be stated.
Conclusions
What implications does it have for clinical practice?

Round 2

Reviewer 1 Report

The one-sided focus on negative narratives is still a problem for the article's trustworthiness. This is a bias in the study that should be accounted for during methodological considerations. The nurses express insecurity in their working environments. The responsibility for the conditions under which they work is placed too much on the individual nurse, and the insecurity is related to their attitudes. What knowledge do you have about their knowledge and attitudes, if you have not asked them about it? In order for the article to be trustworthy, this should be modified, rewritten or other terms used. In addition, it should be explained as the weakness of the study and (as you write) that further research is needed. 

Reviewer 3 Report

The authors have made the changes correctly. Congratulations

Author Response

Dear Reviewer 3

Thank you for your encouragement and helpful comments for improving the quality of this article.